

# Interferometric microscale measurement of refractive index at VIS and IR wavelengths

Meguya Ryu[1], Simonas Varapnickas[2], Darius Gailevicius[2], Domas Paipulas[2],
Eulalia Puig Vilardell[2], Zahra Khajehsaeidimahabadi[3,4], Saulius Juodkazis[2,3,5⋆],
Junko Morikawa[5,6,7†] and Mangirdas Malinauskas[2‡]

**1** National Metrology Institute of Japan (NMIJ),
National Institute of Advanced Industrial Science and Technology (AIST),
Tsukuba Central 3, 1-1-1 Umezono, Tsukuba 305-8563, Japan
**2** Laser Research Center, Physics Faculty, Vilnius University,
Saulėtekio Ave. 10, 10223 Vilnius, Lithuania
**3** Optical Sciences Centre,
ARC Training Centre in Surface Engineering for Advanced Materials (SEAM),
Swinburne University of Technology, Hawthorn, Victoria 3122, Australia
**4** Aerostructures Innovation Research Hub (AIR Hub), Swinburne University of Technology,
John St, Hawthorn, Victoria, Australia
**5** WRH Program International Research Frontiers Initiative (IRFI), Tokyo Institute of
Technology, Nagatsuta-cho, Midori-ku, Yokohama, Kanagawa 226-8503 Japan
**6** LiSM, International Research Frontiers Initiative (IRFI), Tokyo Institute of Technology,
Nagatsuta-cho, Midori-ku, Yokohama, Kanagawa 226-8503 Japan
**7** School of Materials and Chemical Technology, Tokyo Institute of Technology,
Ookayama, Meguro-ku, Tokyo 152-8550 Japan

⋆ sjuodkazis@swin.edu.au , † morikawa.j.aa@m.titech.ac.jp , ‡ mangirdas.malinauskas@ff.vu.lt

## Abstract

**Determination of refractive index of micro-disks of a calcinated (1100°C in air) photo-resist SZ2080™ was carried out using transmission and reflection spectroscopy. Interference fringes at specific wavenumbers/wavelengths were selected for determination of the optical thickness, hence, the refractive index when the thickness of micro-disks was measured by scanning electron microscopy (SEM). Refractive index of disks of $\sim 6 \pm 1$ $\mu$m thickness were determined at visible and IR (2.5-13 $\mu$m) spectral ranges and where $2.2 \pm 0.2$ at visible and IR wavelengths. Peculiarities of optical characterisation of micro-optical structures are discussed in view of possible uncertainties in the definition of geometric parameters, shape and mass density redistribution.**

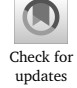

# 1  Introduction

Currently, ultrafast laser assisted 3D micro-/nano-fabrication is used for diverse technological tasks [1–5]. One of the most promising trends is the rapid prototyping and manufacturing of various micro-optical components merging different refractive, diffractive, polarisation, waveguiding and interconnect applications or even combined functionalities [6–10]. They are useful for beam collimation [11], shaping [12], imaging [13], telecommunications [14], sensing [15], hyperspectral imaging [16] and other applications where miniature dimensions are required [17]. Unique virtue of fs-laser pulses in 3D polymerisation is a localised ionization of the major polymerizable component in the photo-resist/resin without the need for photo-initiators [18–21]. Direct write production of 3D silica directly from precursor is a recent example [18–20].

When the wavelength of light becomes comparable to the thickness of an optical element, the wave nature of light manifests stronger as compared with macro-optics [22]. At a larger scale, ray optics and surface curvature of a lens are the major factors for optical function/performance [23]. Characterization of material for applications in micro-optics is preferably based on interference, which has a high sub-wavelength precision in determination of optical thickness - the phase delay [17, 24, 25].

Calcination and sintering of polymerizable materials can provide new composites including glasses, porous glasses polycrystalline and crystalline materials of tailored composition [18, 20, 26–29]. The uniformity of such composite is of paramount importance, similar to the homogeneity of glass, which can only be produced to the high required optical quality for melts in volumes exceeding $\sim 1$ liter. A challenge to produce micro-volumes of composite optical materials with micro-dimensions and good surface optical quality, as well as homogeneous composition and uniform shape, is the next main focus for micro-optics made by fs-laser polymerization or ablation [30,31]. Micro-optics made from polymerizable resists, e.g., SZ2080$^{\text{TM}}$, can be used in IR applications due to small thickness and still acceptable losses due to absorption [32]. Furthermore, SZ2080$^{\text{TM}}$ is a foreseen candidate material for micro-optics and nano-photonics applications due to its compatibility with established functionalization and antireflection coatings for 3D complex multi-layered structures via Atomic Layer Deposition (ALD), which is technology of choice for deposition of conformal layers of tens-of-nm using range of oxides [33]. For such applications, calcination is an interesting next step that could make such optical elements less lossy and sensitive to high laser power irradiation [34]. Typical IR materials CaF$_2$ [35] and BaF$_2$ have melting temperatures 1420° and 1368°, respectively, and are acceptable substrates for calcination up to 1100°C used in this study on SiO$_2$ substrate ($T_m \approx 1710°$C).

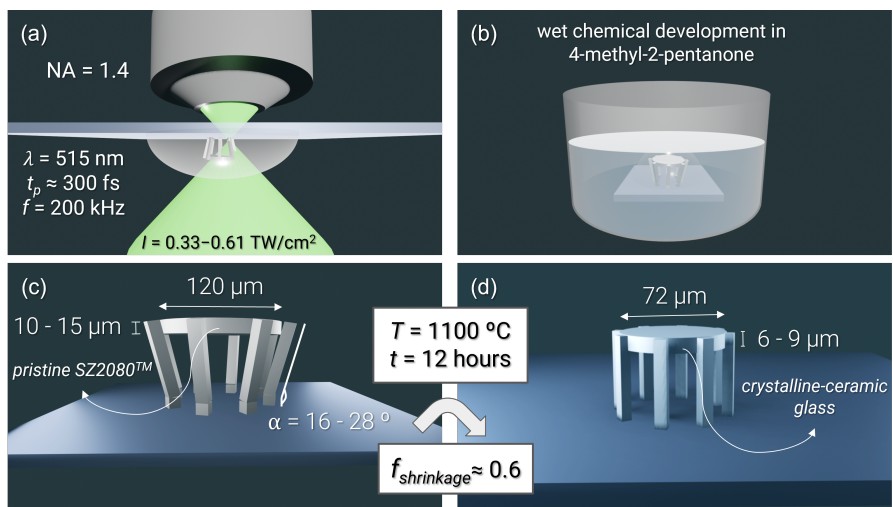

Figure 1: Fabrication protocol of crystalline-ceramic glass 3D micro-optical structures: (a) – selective laser exposure performing multi-photon lithography in pre-polymer SZ2080$^{TM}$; (b) – wet chemical developing revealing the microcture; (c) – obtained 3D structure with shrinkage pre-compensation support for annealing; (d) – final inorganic micro-disc for characterization.

Here, we show the determination of the optical thickness and, consequently, the refractive index at visible (0.4-0.7 $\mu$m) and IR (2.5 -13 $\mu$m) wavelengths of the same $\sim (6-9)-\mu$m-thick slabs of calcinated (1100°C in air) resist SZ2080$^{TM}$ (70-150 $\mu$m in diameter typical for micro-optical elements). We use a route for obtained optical-grade 3D inorganic-structures employing ultrafast laser direct writing multi-photon lithography technique followed by high-temperature annealing. The sequence of the procedure is visually depicted in figure. 1 In figure 2 the concept of this study is shown, where measurements of reflectance and transmittance spectra were used to determine the optical thickness. Such measurements can reveal formation mechanisms and surface tension induced material reflow as well as structural and phase changes induced by calcination when large portion of initial material $\sim$ 40% is removed/oxidised. Having two free surfaces acting onto soft-glass (at high temperature) between them with lateral dimensions much larger than separation is a new unexplored scenario of micro-optics formation.

## 2 Experimental

Micro-disks of the final thickness of $d = 7 \pm 2$ $\mu$m after calcination of SZ2080$^{TM}$ with 1% w.t. Irgacure 369 as photoinitiator were fabricated at different exposure doses using fs-laser (Yb: KGW Pharos, Light Conversion) at average power range $P_{av} = [70-130]$ $\mu$W (corresponding to estimated intensity at focus $I = 0.33 - 0.61$ TW/cm$^2$); the wavelength $\lambda = 515$ nm, pulse duration $t_p \approx 300$ fs repetition rate $f = 200$ kHz. The system comprises a galvanometric beam scan system with precision translation stages [36], which can work synchronously ensuring continuous 3D writing without causing any stitching [37]. The fabrication process was controlled by 3DPoli (Femtika) software at a linear stage scan at high speed $v_{sc} = 10$ mm/s, slicing between layers along the optical axis at the step of $\Delta z = 0.2$ $\mu$m, in-plane hatching (radial concentric) was $\Delta r = 50$ nm and corresponded to the in-line separation between neighboring pulses $v_{sc}/f = 50$ nm. Such a condition corresponds to the most uniform exposure of the focal volume. A Plan Apochromat Zeiss 63$^x$ magnification and with numerical aperture $NA = 1.4$

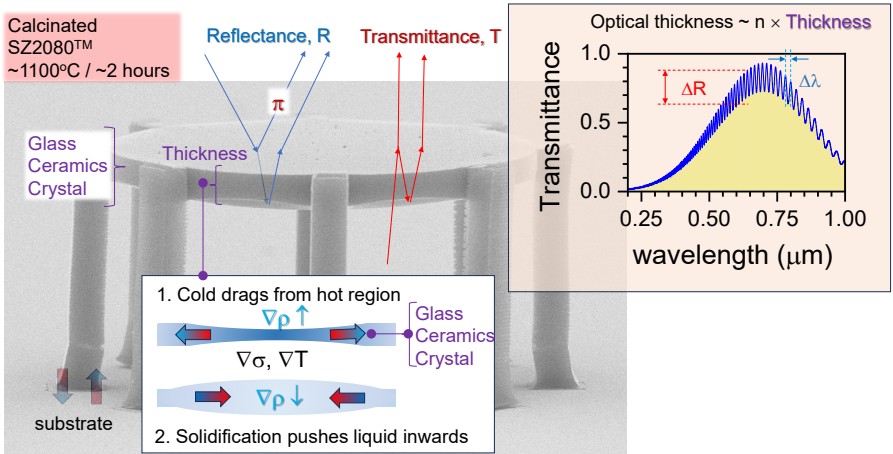

Figure 2: What is the refractive index $n \propto \rho$ ($\rho$ is the mass density) of the calcinated micro-disks out of SZ2080$^{\text{TM}}$ resist? Concept of the study: spectra of reflectance $R(\lambda)$ and transmittance $T(\lambda)$ are used to determine the optical thickness $n(\lambda) \times d$ at visible and IR spectral domains; $d$ is the thickness, $\lambda$ is the wavelength. Surface tension $\sigma(T, r)$ driven flows on *two* surfaces are defined by the gradients $\nabla \sigma$ caused by the local temperature gradient $\nabla T$ varying over position $r$. Different scenarios 1 and 2 of the $\sigma$-induced mass transport can cause increase or decrease of $n$. Glass transition at specific temperature, phase separation due to exsolution, and crystalisation all can contribute to the final refractive index and thickness of the calcinated micro-optical element. Concept of measurements (right-inset): the transmittance (or reflectance) spectrum is modulated $\Delta R$ ($\Delta T$) with a measurable wavelength spacing $\Delta \lambda$ fringes due to reflections in a Fabri-Perot slab (as depicted in SEM image of slab-disk).

immersion oil objective lens was employed to focus the laser beam inside the SZ2080$^{\text{TM}}$ resist through an oil immersion; resist was on fused silica glass substrates. It is known that during calcination SZ2080$^{\text{TM}}$ structures shrink by a factor of 0.6 [26] and this was taken into account by making them appropriately larger during initial fabrication. Too small of a structure after calicnation would have made their characterization more difficult, therefore the primary design was set to a 120 $\mu$m diameter disk, expected to become 72 $\mu$m, close to experimental observation $\sim 70$ $\mu$m.

The laser power of 100 $\mu$W corresponds to $I_p = 0.474$ TW/cm$^2$ per pulse; this condition was used to polymerize micro-lenses directly on a glass substrate without calcination [22]. The diameter at focal spot was $2r = 1.22\lambda/NA = 445$ nm. The support pillars were recorded in raster scan mode with layer-by-layer separation of $\sim 0.57$ $\mu$m after calcination, which closely corresponds to the designed 1 $\mu$m step before calcination (0.57 $\mu$m is 60% of the initial size, 100% corresponds to 0.95 $\mu$m).

In previous study, the required geometry was established to pre-compensate shape of polymerised posts supporting the micro-optical element in order to retrieve the final optical element with close-to-normal support pillars after calcination [38].

Figure 3 shows optical and SEM images of the calcinated micro-disks. Inset of Fig. 3(a) shows the structure made at the lowest power. Slight changes of surface-plane of micro-disks with substrate could take place as well as slight thickness $d$ variations, which were the focus of the analysis discussed below. Micro-disks were suspended above fused silica substrate by 35-37 $\mu$m. After calcination they were very close to the circular circumference as fs-laser polymerised, only reduced in size by $\sim 40\%$ due to calcination during which the organic part

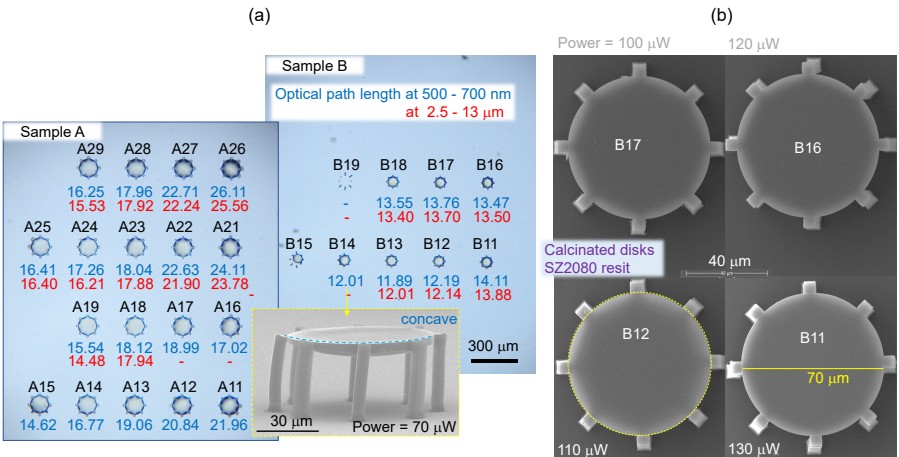

Figure 3: Calcinated micro-disks out of SZ2080$^{\text{TM}}$. (a) Optical images of samples polymerised at different laser powers; the determined optical thickness $nd$ at the visible 500-700 nm window and IR at 2.5-13 $\mu$m is shown; the sale bar for both optical images is the same. Inset shows a side-view SEM image of a suspended B14 micro-disk at the lowest power when the disk acquired a concave shape recognizable on the top plane. (b) SEM top-view images of the calcinated B-sample (smaller) micro-disks.

of the resist was fully removed (oxidised) at high temperature of 1100°C for 12 hours [38]. Uniform resizing of polymerized structures during calcination is the virtue of this method [26].

There is a possibility of slight changes of the surface inclination of micro-disks during calcination. SEM imaging was used (Fig. 4) to determine the angle (normal to the surface of silica substrate, which is 0° for the top-view imaging in SEM). With the $\theta$ angle determined from the elliptical profile of circular disks (Fig. 4), the thickness $d$ of micro-disks was determined for calculations of refractive index $n_2$ of the calcinated disk from the optical thickness $n_2 d$.

The optical thickness $n_2 d$ was determined from the interferometric fringe structure in both IR refraction and the visible transmission spectra. IR reflection spectra were measured by IR imaging system (Spotlight, Perkinelmer) with the pixel size of 6.25 $\mu$m × 6.25 $\mu$m. The wavenumber resolution of the spectra measurements were set to 4 cm$^{-1}$ and the spectrum range was set between 4000 cm$^{-1}$ and 720 cm$^{-1}$. The visible transmission spectra were measured by Czerny-Turner type monochromator (OceanOptics, Flame-S). The mnochromator and the microscope (Olympus, BX43) were connected by optical fiber ($\phi = 400$ $\mu$m). The width of the slit at the input port of the monochromator was 25 $\mu$m, resulting 1.33 nm wavelength resolution (FWHM).

## 3 Methods

### 3.1 Poly-chromatic interferometry

The optical path length of the sample was measured by the poly-chromatic interferometry in the reflection or transmission geometry. When a normally incident polychromatic plane wave transmits a thin transparent layer (sample) of refractive index $n_2$ and thickness $d$ surrounded by transparent media having lower refractive index $n_1$ such as air, the amplitude of the transmitted light after several inter-reflection in the gap will be a series of complex amplitudes

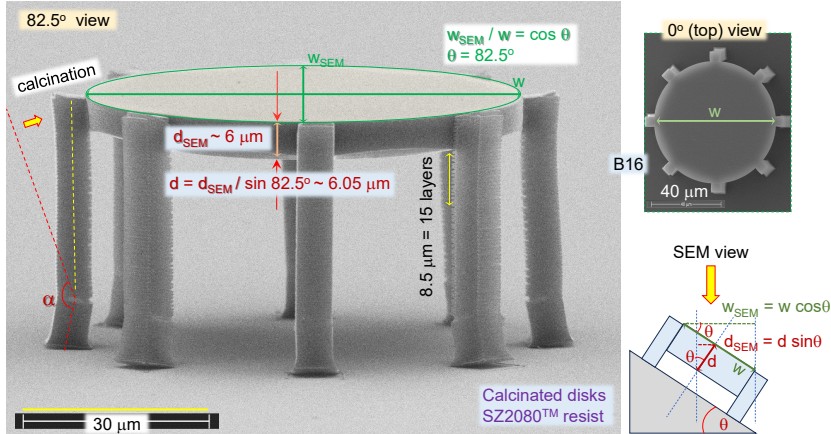

Figure 4: Actual tilt angle $\theta$ of SEM imaging was determined from the ellipticity of imaged circular micro-disk. The entire structure was polymerised out of SZ2080$^{TM}$ with support pillars at angle, which was straightened towards a vertical orientation after calcination. The average fs-laser power was 120 $\mu$W. Separation of layer-to-layer for the support pillars was $567 \pm 20$ nm. Thickness of a disk was evaluated at the same location on the structure as shown here for all the samples from the same SEM session (same tilt of the sample).

$E_1$, $E_2$, $E_3$,..., expressed as follows (the Stokes' treatment [39]):

$$E_1 = E_i t t',$$

$$E_2 = E_i t t' r^2 e^{i(2\phi+\delta)}, \tag{1}$$

$$E_3 = E_i t t' r^4 e^{i(4\phi+2\delta)} \ldots \tag{2}$$

Here, $E_i$ is the incident light amplitude, $r$ is the reflection coefficient at the surface of the sample, $\phi$ is the phase change associated with a single reflection, $\delta = 2\frac{2\pi}{\lambda}nd$ is the phase change due to the optical path difference, and $t$ and $t'$ are the transmission amplitude coefficient for the substrate-sample and sample-substrate interface, respectively. By neglecting terms in $r$ of higher order than 4, transmitted light amplitude $E_t$ can be written as follows:

$$E_t = E_i t t' \left(1 + r^2 e^{i(2\phi+\delta)}\right). \tag{3}$$

The intensity $I_T = E_t \cdot E_t^*$ can be written as:

$$I_T = E_t \cdot E_t^* = E_i \cdot E_i^* t^2 t'^2 \left[1 + r^2 \left(e^{i(2\phi+\delta)} + e^{-i(2\phi+\delta)}\right) + r^4\right], \tag{4}$$

and, by using the Euler's formula $\left(e^{i\delta} + e^{-i\delta}\right) = 2\cos\delta$, the equation can be further simplified:

$$I_T = I_{in} t^2 t'^2 \left[1 + 2r^2 \cos\left(\frac{4\pi nd}{\lambda}\right) + r^4\right] = I_{in}(1-r^2)^2 \left[1 + 2r^2 \cos\left(\frac{4\pi nd}{\lambda}\right) + r^4\right], \tag{5}$$

here identify $t t' = 1 - r^2$ is used; $r^2$ is the portion of intensity reflected by one interface. Transmitted intensity is complimentary to reflection at negligible absorption $I_T + I_R = 1$. This Eqn. 5 was validated to fit experimental results of low-temperature 300°C calcinated micro-disks with prisms (see Fig. 5). To account for the actual dispersion of sample transmittance,

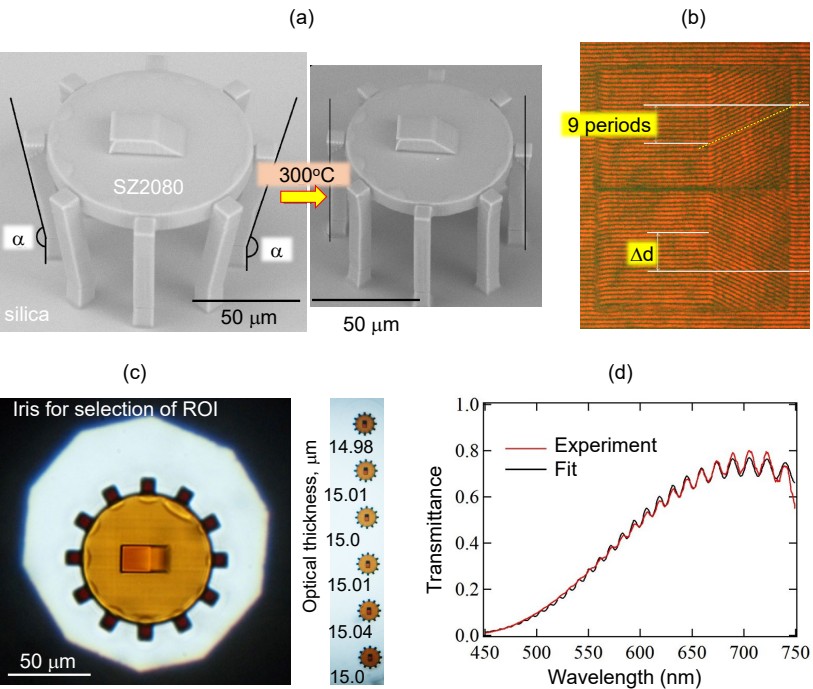

Figure 5: Method. Interferometric determination of optical thickness $n_2d$. (a) SEM images of polymerised and calcinated (at 300°C) SZ2080 structures. (b) Interferometric transmission imaging of a calcinated prism using $\lambda = 632$ nm HeNe laser illumination of a sample and image formed with Michelson interferometer. The fringes can be traced to determine the thickness of the prism which corresponded to optical thickness of $n_2d = n_2 \times 9(\lambda/n_2) = 5.688$ $\mu$m. (c) Transmission imaging with iris selection of ROI. Inset shows optical images of calcinated structures and the optical thickness $n_2d$ measured from selected ROI and fitting $T$ spectra by Eqn. 6 as shown in (d). Transmittance spectra $T(\lambda)$ from ROI (c). Calcination at lower temperature and shorter time yielded in yellow-brown tint of the modified SZ2080$^{\text{TM}}$.

the fit function was chosen with a Gaussian preform multiplier to reproduce the spectral form factor:

$$T(\tilde{v}) = a_G \exp\left(-\frac{(\tilde{v}-b_G)^2}{2c_G^2}\right) \times [1 + a_t \sin(2\pi b_t \tilde{v} + c_t)]. \tag{6}$$

Here $a_G, b_G, c_G$ are the Gaussian form factor parameters to reproduced spectral form factor, $\tilde{v} = 1/\lambda$ is the wavenumber, $b_t = 2n_2d$ is the optical thickness, which is the major fit parameter, $a_t$ is related to $r^2$ and $c_t$ is the phase angle of the fit (a constant related to the Fresnel coefficients and anisotropies of $n$ and $\kappa$). The fit is shown in Fig. 5(d). This procedure to determine $n_2d$ was used for transparent micro-disks calcinated at high 1100°C temperature; figure 5(d) illustrates the method's applicability to similar samples calcinated at 300°C.

It is noteworthy, that only polymerised part of the structure was contributing to the formation of interference fringes (see images in Fig. 5(b)) while the air-gap between the structure and substrate did not contribute to the modulation. This can be achieved by conditions of focusing (*NA*, placement of the focal region) and was always checked experimentally for the $T$ and $R$.

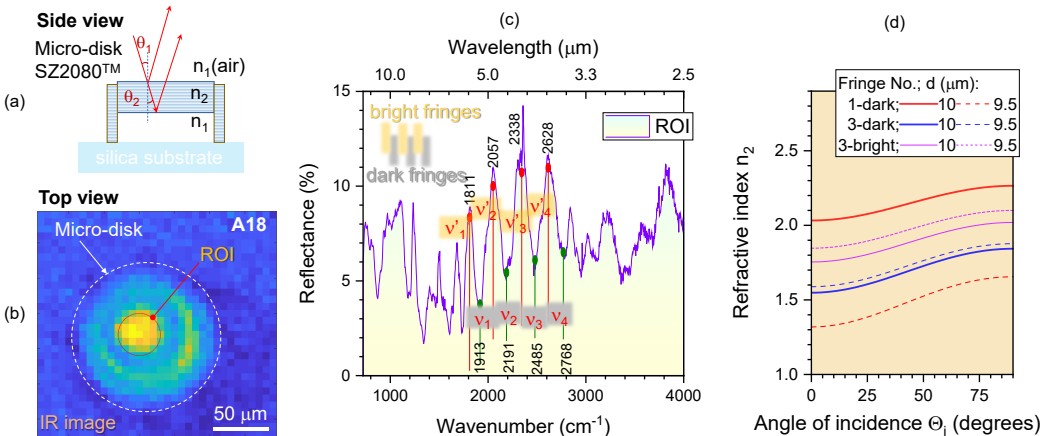

Figure 6: (a) Principle of interference method in reflection at angle of incidence $\theta_i$. (b) IR image of a micro-disk. (c) IR spectrum measured from the region of interest in (b); markers are in wavenumbers $\tilde{\nu}$. The wavenumbers $\tilde{\nu}_i = \frac{1}{\lambda_i}$ at reflection maxima used for calculation by Eqn. 9. Normalisation of reflectance $R$ was carried out using Au mirror. (d) Visualisation of Eqn. 9 for two thickness settings of 10 and 9.5 $\mu$m determined from two adjacent peaks (one dark fringe) $\tilde{\nu}_{1-2}$, $\tilde{\nu}_{1-4}$ (3 dark fringes) and from minima $\tilde{\nu}'_{1-4}$ (3 bright fringes); the window of analysis $\lambda = 3 - 5$ $\mu$m.

## 3.2 Reflectance

Reflection from a medium with a higher refractive index $n_2$ for light incident from the medium of lower refraction index $n_1$ (air) at an incident angle $\theta_i$ is used to measure optical thickness $n_2 d$ of the dielectric slab (micro-disk in this study). Importantly, the medium below the sample has a lower refractive index $n_3 < n_2$; this is required since there would be a $\pi$ phase change for reflected light at the boundary 3-to-2 if $n_3 > n_2$ as for the front surface 1-to-2 interface (marked in Fig. 2).

Interference maxima and minima in reflected spectra from a slab/film of thickness $d$ are determined by the conditions for the phase optical thickness for a double pass in the slab $2nd \cos\theta_2$ [40]:

$$2nd \cos\theta_2 = m/\tilde{\nu} \quad \text{(Minima)}, \tag{7}$$

$$2nd \cos\theta_2 = \left(m + \frac{1}{2}\right)/\tilde{\nu} \quad \text{(Maxima)}, \tag{8}$$

where $\theta_2$ is the refraction angle in the slab/film (medium 2), $m$ is the order of interference, and $\tilde{\nu} = 1/\lambda$ is the wavenumber ($\lambda$ is the wavelength). From Snell's law we have $\cos\theta_2 = \sqrt{1 - \sin^2\theta_1/n_2^2}$. By combining this expression with conditions of either minima or maxima (Eqn. 7;8) one finds [40]

$$n_2 = \sqrt{\frac{(\Delta m)^2}{4d^2(\Delta\tilde{\nu}_{if})^2} + \sin^2\theta_1}, \tag{9}$$

where $\Delta m = m_i - m_f$ is the number of fringes between the initial and final fringes counted, and $\Delta\tilde{\nu}_{if} = \tilde{\nu}_i - \tilde{\nu}_f$ is the difference between the corresponding fringes. This Eqn. 9 was used to determine the refractive index at IR spectral ranges at close to normal incidence $\theta_1 \approx 0°$. Slight changes of $\theta_1$ due to sample tilt can be modeled using Eqn. 9.

# 4 Results and discussion

Figure 6 shows the principle of measurement of refractive index from optical thickness in reflection (a). Equation 6 was used for the reflection spectra at normal incidence, hence $\theta_i \equiv \theta_1 = 0°$. The IR image of the micro-disk, which is close in thickness to the IR wavelengths is shown in (b) and the corresponding IR spectra in (c). By selecting several orders of interference maxima, a more precise determination of refractive index $n_2$ can be made (assuming negligible dispersion within that range). Using the initial and final wavenumbers $\tilde{\nu}_{i,f}$, Eqn. 9 was used to determine $n_2$ from experimentally determined $d$ from the rim of the disk imaged at large tilt angle $\theta \sim 82°$ (Fig. 4). The method is sensitive to the definition of spectral positions of the maxima as illustrated in the inset of (c). If only two adjacent maxima (one fringe [40]) are selected, there is a large difference in thickness determination when it is changing by only 5% (10 vs. 9.5 $\mu$m) as shown in the inset of Fig. 6(c). If four peaks (three fringes) are used, the difference in the thickness determination is much smaller regardless minima (bright fringes) or maxima (dark fringes) are used. Also, the $n_2$ values are strongly different for the 1 vs. 3 fringes. Selection of fringes was made at the spectral window where narrow IR absorption bands of SZ2080$^{\text{TM}}$ are not pronounced [32].

It is noteworthy, that the second term of $\sin^2 \theta_1$ in Eqn. 9 can affect the determined $n_2$ value. This is of particular importance for structures that might lose alignment of the surface plane of the disk with the substrate after calcination (see, inset in Fig. 3(a)). The center-line of oscillating reflectance spectra is close to the expected value of $n = 1.7$ as evident from normal incidence (from air) $R = \frac{(n_2-1)^2}{(n_2+1)^2} = 6.7\%$, which was close to experimentally observed $R$ values.

Poly-chromatic interferometry (Sec. 3.1) was used for the determination of optical thickness $n_2 d$ (and $n_2$ for the measured $d$) at visible spectral range using the white light illumination of a microscope. The region of interest (ROI) was selected over the micro-disk (see the condition in Fig. 5(c)) and transmitted spectra was fitted by Eqn. 6 as shown in Fig. 5(d). Fitting over the spectral range of 500-700 nm provided high fidelity of fitting due to many periods of oscillating spectra. Figure 7 has a summary of results of $n_2 d$ determination at visible and IR spectral wavelengths made by two complementary methods of polychromatic interferometry (visible) and reflectivity (IR) for the same calcinated micro-disks. Closely matching $n_2 d$ values were obtained for the $0.5-0.7$ $\mu$m as well as $3-5$ $\mu$m ranges. Interestingly, thinner disks polymerised at lower powers were easily characterised at IR wavelengths due to low absorbance, however, at visible wavelengths, the consistent fringe pattern was not always observed, most probably, due to wedge formation, concave center, etc. (see, scenarios 1 and 2 in Fig. 2) which affect shorter wavelengths considerably stronger.

For the mid-range power of 100 $\mu$W polymerisation, the difference in optical thickness is from $n_2 d = 13 \pm 1$ $\mu$m (sample B) with $d_{SEM} = 6$ $\mu$m with $n = n_2 d / d_{SEM} = 2.17$. Larger optical thickness $n_2 d = 19 \pm 1$ $\mu$m was determined for the approximately twice larger microdisks (two regions on sample A). The difference of $\sim 32\%$ in optical thickness was not expected due to the same laser exposure protocol and a total dose. A slight difference in sample's preparation and development could be the reason, which led to different material photo-sensitivity, degree of cross-linking, and density [24, 25] – all further influencing the evolution of calcination. Smaller samples (B) tend to be over-developed since it is judged by a naked-eye observation of 3D structures on glass. Over-exposed samples are more nano-porous on the surface which helps the removal of organic components during calcination. Larger disks were slightly thicker and less porous, which led to a slightly different evolution of calcination yielding disks of a larger retardnce and larger thickness $\frac{n_2 d}{d_{SEM}} = \frac{19 \, \mu\text{m}}{8.5 \, \mu\text{m}} \approx 2.24$.

Both, $n_2$ and $d$ can be affected by calcination at high temperatures due to compositional changes of glass-ceramic (silica-zirconia). In the case of suspended micro-opitcs (disks here), the surface tension-driven mass transport (Fig. 2) of lower viscosity regions (higher temper-

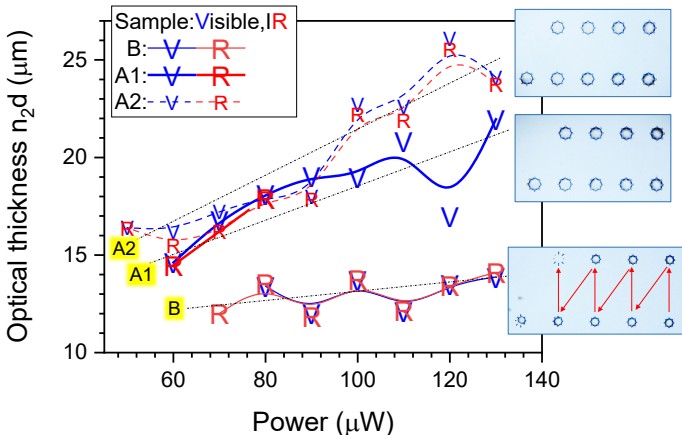

Figure 7: Optical thickness $n_2 d$ of three samples: B, A1, A2 measured in transmission for the visible spectral range (Eqn. 6) and in reflection for IR (Eqn. 9) vs. fs-laser power. Lines are showing a global trend as an eyeguide for the same sample. Insets show optical images of arrays of micro-disks all at the same scale . Laser polymerisation at average power of 100 $\mu$W corresponded to $I_p = 0.474$ TW/cm$^2$ per pulse at $f = 0.2$ MHz laser repetition rate; the focusing was carried out with $NA = 1.4$ objective lens. Arrows in the inset show the sequence of decreasing powers used in laser printing of micro-disks.

ature) can also contribute to the final state of processing [41]. At temperatures above the annealing point of glass, the surface tension is changing by $4 \times 10^{-5}$ N/m per 1K degree for typical $\sigma = 0.23 - 0.36$ N/m for different glasses [42].

As shown above, the determination of $n_2$ is strongly dependent on the measurement of micro-slab thickness $d$, which was carried out using a tilted view in SEM (Fig. 4). Higher power of fs-laser during 3D polymerisation, hatching, scan speed at selected repetition rate and numerical aperture of the focusing lens all contributed to the 3D accumulated exposure dose, hence, to the final cross-linking, mass density and, consequently, refractive index $n_2$. This is evident from Fig. 7. Mass redistribution inside micro-disks during thermal cooling stage could be expected with gradients of less viscous glass phase (higher local temperature) pulled towards the colder regions (larger surface tension), e.g., support pillars [41]. Thermal morphing of laser processed 3D structures made out of glass was demonstrated by changing shape of 3D object cube-to-sphere depending a temperature-time protocol [43]. Since the micro-disks are suspended, two surfaces can affect the liquid glass between two colder surfaces, which is very different from the known cases of observed mass redistribution by re-solidification front on a molten phase after laser ablation (one surface with air). Glasses are known for the low atomic packing $C_g$ and can experience significant densification under pressure [44]. For amorphous silica, Poisson's ratio $v = 0.15$ is low and is correlated with $C_g$ as it is typical for glasses [44]. Hence a lateral compaction is not causing significant out-off-plane expansion due to low Poisson's ratio, however, densification can occur (a larger refractive index). Interesting opportunities in metal oxide reduction and mixing could be explored using exsolution at high temperatures using polymerised materials during the calcination step [45].

It is noteworthy that homogeneity of the shape of a micro-disk, as well as interferometric transmission measurements, showed that the entire thickness of the calcinated small 3D prisms (at 300°C) have the same mass density distribution (same pattern of interference fringes; Fig. 5(b)). The higher refractive index of a silica-zirconia calcinated resist is expected due to very different melting temperatures of SiO$_2$ at 1710°C, SiO at 1705°C and zirconia at 2750°C;

silica sublimation rate at 1700K is $\sim 10^{-4}$ g/cm$^2$/s [46]. Partial loss of silica as compared to zirconia could contribute to the increase of refractive index of calcinated micro-optics. The variation in exact refractive index value could be influenced by environmental conditions such as ambient humidity.

## 5 Conclusion and outlook

Calcination of micro-disks at high 1100°C for 12 hours in air resulted in high refractive index slabs of $n \approx 2.2 \pm 0.2$ at visible and IR (2.5-13 $\mu$m) wavelengths. Transmission (visible) and reflection (IR) modes of spectroscopic determination of optical thickness was closely matched each other. However, there was strong heterogeneity of optical thickness between disks which had twice different diameters and were developed for different times (longer development for smaller 3D structures). Calcination evolution was affected by the development and surface nano-porosity. This calls for a dedicated investigation to establish reliable protocols for defined refractive index, which can be also affected by structural and material changes [47]. The increase of refractive index from 1.6 (visible) in a polymerised state can be related to the partial sublimation of silica and enrichment of calcinated material by zirconia. Also, densification of composite during calcination by two free surfaces (Fig. 2) open new avenues for better compaction of glassy composite which usually has a low atomic packing. The low Poisson's ratio of glasses facilitates their predictable size rescaling during calcination. Creation of composition-tailored glasses and glass-ceramics via calcination is promising for realization of high durability micro-optical [48] components suitable for intense light non-linear microscopy applications [49] as well operational in diverse harsh environments [50].

## Acknowledgments

**Funding information**   D.G, D.P., E.P.V., S.J. and M.M. acknowledge the "Universities' Excellence Initiative" programme by the Ministry of Education, Science and Sports of the Republic of Lithuania under the agreement with the Research Council of Lithuania (project No. S-A-UEI-23-6). E.P.V. participated as part of her master thesis project in the EuroPhotonics (International Master in Photonics program) scholarship. S.J. was supported by the DP240103231 grant from Australian Research Council. Z.K. is supported by the Aerostructures Innovation Research (AIR) Hub at Swinburne University of Technology for this research project and for PhD scholarship. M.R. and J.M. were partially supported by JST CREST Grant Number JP-MJCR19I3, Japan.

## A Transmittance

The very same method presented above for the reflectance spectra $R(\tilde{\nu})$ can be applied for the transmittance $T(\lambda)$ with the exchange of the minima and maxima conditions as one would expect for the optically thin slab with $T = 1 - R$ (for negligible absorbance $A \to 0$) [51]:

$$2nd \cos\theta_2 = m\lambda \quad \text{(Maxima)}, \tag{A.1}$$

$$2nd \cos\theta_2 = \left(m + \frac{1}{2}\right)\lambda \quad \text{(Minima)}. \tag{A.2}$$



The optical losses due to absorption through a slab of thickness $d$ are accounted by [52]

$$T(\lambda) = \frac{(1-R(\lambda))^2 e^{-\alpha(\lambda)d}}{1-R^2(\lambda)e^{-2\alpha(\lambda)d}} I_{in}(\lambda), \tag{A.3}$$

where $I_{in}$ is the intensity of incident light, $\alpha(\lambda) = 4\pi\kappa/\lambda \ [\text{cm}^{-1}]$ is the absorption coefficient; the second term in denominator become negligible only for the strong absorption $\alpha d \geq 1$.

Since $r^2 < 1$ (Eqn. 1), it is possible to calculate all the infinite sum of the transmitted fields (Eqn. 3) [39]:

$$I_T = I_{in} \frac{(1-r^2)^2}{1-2r^2\cos\delta + r^4} \equiv \frac{I_{in}}{1+[4r^2/(1-r^2)^2]\sin^2(\delta/2)}. \tag{A.4}$$

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
