# Peer review of "Interferometric microscale measurement of refractive index at VIS and IR wavelengths"

_SciPost Physics, doi:SciPost Phys. Core 7, 059 (2024)_

## Round 1 · Referee Report · Anonymous (Referee 1) · 2024-4-1

Report

While it may be note groundbreaking (per SciPost Physics first criteria) results, the authors present a solid and scientifically sound study on the optical properties of microfabricated disks made from photoresist. The refractive index has been determined on microscales using a conventional approach based on interference fringes in spectral response. This method, known and robust, has been demonstrated by the authors to be applicable at such scales and for this particular purpose.

Major comment:
It is plausible that the authors have addressed this issue in the manuscript, but it wasn't readily apparent to me. The authors assumed interference solely from reflections off both surfaces of the microdisk, which are approximately 6 micrometers apart. However, a similar interference effect may arise from the reflection of the beam off the microdisk and the fused silica substrate. The separation between microdisk and substrate of 40 um and the refractive index of fused silica should induce a comparable level of interference modulation, which should indeed be present. This is similar to etalon. Furthermore, similar to the described beams, such interference might be observed in both reflection and transmission. Is it possible that the observed fringes result not solely from the interference of reflections from the microdisk surfaces (akin to thin film interference) but also from the microdisk substrate? The Newton ring-like pattern (Figure 5b) also supports this notion. This may significantly change modelling and results.

Minor:
1. This is a subjective opinion, but it would be beneficial to juxtapose reflection and transmission data to illustrate the correlation in the interference pattern. Currently, the primary focus seems to be on reflection alone. Two graphs put together can be very useful.
2. Again, this is subjective. While Equation 9 is undoubtedly correct, the message surrounding it and Figure 5d is somewhat confusing. It intuitively suggests that the refractive index is dependent on the incident angle (which is a fundamental property of materials), whereas the real significance lies in the importance of knowing the incident angle with high precision to determine the refractive index from the data. Perhaps some rewording could clarify the uncertainty and the dependence on the incident angle.

---

## Round 1 · Referee Report · Anonymous (Referee 2) · 2024-4-15

Report

The authors present the fabrication of the micro-fabricated photoresist disks and the polychromatic interferometric determination of the optical retardation. It seems the fabricated structures have similar retardation in a wide energy range (in visible and IR ranges) and it is scaled with the disk size (at least for the two sizes that are mainly discussed. I think the authors managed to fabricate very homogeneous disks with homogeneous qualities and characterize their disks with rather standard interferometric measurements. The overall work is interesting, but I fail to see the groundbreaking aspect of it, especially when there is still room for improvement in the measurement and analysis aspects. Therefore, I think it will be more suitable for the SciPost Physics Core.

Here are some of the points that authors should consider for the revision:
1. Although the interferometric determination of the optical properties might be rather standard, no details about the measurement setup is given. Perhaps for the general reader, it could be helpful to describe the experiments leading to the transmission and reflection spectra.

2. Authors describe their spectra with only a single slab problem assuming that the disk is sandwiched in between air. However, I think the situation is more complex than this, as there should be multiple reflections between the sample and the silica substrate, as well as the substrate itself (especially in the IR region, where the silica is only partially transparent, at least in some portion of the measurement range). The situation is also visible in the reflectance curve in Fig.5, where one can see multiple shoulders to the peaks, etc. I think the correct determination of the peak positions is also important for the correct determination of the refractive index.

Minor comments:
a. It should be Michelson interferometer (in Fig.4 it is given as Mickelson interferometer!)

b. Sometimes the information on the figures are not described (or it is not immediately visible) for instance, in Fig.1, Transmittance spectrum is given with green and red dashed lines defining something, which is not very clear what. In Fig. 2, Sample A and B are given but their difference becomes clear only after careful reading and it is not immediate. I realize that the authors put the scale, but it feels like only belongs to Sample B.

c. In the text, authors mentioned that they fit the Transmittance with Eq.5 (Last paragraph of page 8), while in the Figure4 caption, it is mentioned that it was Eq.6 (which is probably the one).

Recommendation

Accept in alternative Journal (see Report)

---

## Round 2 · Author Response

List of changes

Answer. Thank you for the very good questions. Indeed, this was issue we checked thoroughly. Among different fabricated samples there were structures with tilted disk in respect to the substrate as well as slightly different disk-substrate distances. We looked for the difference in fringe pattern as well as made numerical predictions what periodicity to expect in the measyred reflection spectra. This allowed us to exclude the possibility of those resonant features to change and influence current interpretation. Revisions were made to better reflect this.
Minor:
1. This is a subjective opinion, but it would be beneficial to juxtapose reflection and transmission data to illustrate the correlation in the interference pattern. Currently, the primary focus seems to be on reflection alone. Two graphs put together can be very useful.
Answer. Good point. It was not possible to measure both R and T spectra from the same point in the used setup due to mechanical constraints of the condenser and objective positions and scanning range. Since reflection is always a simpler method and can be applied at visible and IR wavelength for thin and absorbing samples we focused on R measurements. The appendix shows all formulae applicable to transmittance.
2. Again, this is subjective. While Equation 9 is undoubtedly correct, the message surrounding it and Figure 5d is somewhat confusing. It intuitively suggests that the refractive index is dependent on the incident angle (which is a fundamental property of materials), whereas the real significance lies in the importance of knowing the incident angle with high precision to determine the refractive index from the data. Perhaps some rewording could clarify the uncertainty and the dependence on the incident angle.
Answer. Figure 5d shows measured spectra from the selected region by an aperture. The interference pattern is caused by the polymerized membrane, which the actual form factor of the spectra is also affected by rest of objects and structures inside selected region (ROI). This is why, the fit is taken to make qualitative match, while the main information is in the spectral positions of the peaks. Rewording is made to better reflect this.

1. Although the interferometric determination of the optical properties might be rather standard, no details about the measurement setup is given. Perhaps for the general reader, it could be helpful to describe the experiments leading to the transmission and reflection spectra.
Answer. Thank you for the remark. Description added.
Authors describe their spectra with only a single slab problem assuming that the disk is sandwiched in between air. However, I think the situation is more complex than this, as there should be multiple reflections between the sample and the silica substrate, as well as the substrate itself (especially in the IR region, where the silica is only partially transparent, at least in some portion of the measurement range). The situation is also visible in the reflectance curve in Fig.5, where one can see multiple shoulders to the peaks, etc. I think the correct determination of the peak positions is also important for the correct determination of the refractive index.
Answer. Good point. We measured several samples with different geometries with inclined polymerised membrane also when membranes were at different heights. This was compared with expected Fabry-Perot fringes. This analysis allowed us to make assignments we use in the description. Shoulders in the IR spectra are caused by absorption rather competing resonators. Description improved.
Minor comments:
a. It should be Michelson interferometer (in Fig.4 it is given as Mickelson interferometer!)
Answer. Corrected. Thank you.
b. Sometimes the information on the figures are not described (or it is not immediately visible) for instance, in Fig.1, Transmittance spectrum is given with green and red dashed lines defining something, which is not very clear what. In Fig. 2, Sample A and B are given but their difference becomes clear only after careful reading and it is not immediate. I realize that the authors put the scale, but it feels like only belongs to Sample B.
Answer. Thank you. Improved.
c. In the text, authors mentioned that they fit the Transmittance with Eq.5 (Last paragraph of page 8), while in the Figure4 caption, it is mentioned that it was Eq.6 (which is probably the one).
Answer. Thank you for showing the overlooked mistake. Corrected.
* * *
Extra Fig 1 was created to introduce into the fabrication protocol of the samples. Additional text marked in blue was inserted in Introduction section: “We use a route for obtained optical-grade 3D inorganic-structures employing ultrafast laser direct writing multi-photon lithography technique followed by high-temperature annealing. The sequence of the procedure is visually depicted inf Figure. 1”. We believe it gives more clarity regarding the motivation and preparation of the 3D crystalline-ceramics glass microstructures.

---

## Round 2 · List of Changes

Answer. Thank you for the very good questions. Indeed, this was issue we checked thoroughly. Among different fabricated samples there were structures with tilted disk in respect to the substrate as well as slightly different disk-substrate distances. We looked for the difference in fringe pattern as well as made numerical predictions what periodicity to expect in the measyred reflection spectra. This allowed us to exclude the possibility of those resonant features to change and influence current interpretation. Revisions were made to better reflect this.
Minor:
1. This is a subjective opinion, but it would be beneficial to juxtapose reflection and transmission data to illustrate the correlation in the interference pattern. Currently, the primary focus seems to be on reflection alone. Two graphs put together can be very useful.
Answer. Good point. It was not possible to measure both R and T spectra from the same point in the used setup due to mechanical constraints of the condenser and objective positions and scanning range. Since reflection is always a simpler method and can be applied at visible and IR wavelength for thin and absorbing samples we focused on R measurements. The appendix shows all formulae applicable to transmittance.
2. Again, this is subjective. While Equation 9 is undoubtedly correct, the message surrounding it and Figure 5d is somewhat confusing. It intuitively suggests that the refractive index is dependent on the incident angle (which is a fundamental property of materials), whereas the real significance lies in the importance of knowing the incident angle with high precision to determine the refractive index from the data. Perhaps some rewording could clarify the uncertainty and the dependence on the incident angle.
Answer. Figure 5d shows measured spectra from the selected region by an aperture. The interference pattern is caused by the polymerized membrane, which the actual form factor of the spectra is also affected by rest of objects and structures inside selected region (ROI). This is why, the fit is taken to make qualitative match, while the main information is in the spectral positions of the peaks. Rewording is made to better reflect this.

1. Although the interferometric determination of the optical properties might be rather standard, no details about the measurement setup is given. Perhaps for the general reader, it could be helpful to describe the experiments leading to the transmission and reflection spectra.
Answer. Thank you for the remark. Description added.
Authors describe their spectra with only a single slab problem assuming that the disk is sandwiched in between air. However, I think the situation is more complex than this, as there should be multiple reflections between the sample and the silica substrate, as well as the substrate itself (especially in the IR region, where the silica is only partially transparent, at least in some portion of the measurement range). The situation is also visible in the reflectance curve in Fig.5, where one can see multiple shoulders to the peaks, etc. I think the correct determination of the peak positions is also important for the correct determination of the refractive index.
Answer. Good point. We measured several samples with different geometries with inclined polymerised membrane also when membranes were at different heights. This was compared with expected Fabry-Perot fringes. This analysis allowed us to make assignments we use in the description. Shoulders in the IR spectra are caused by absorption rather competing resonators. Description improved.
Minor comments:
a. It should be Michelson interferometer (in Fig.4 it is given as Mickelson interferometer!)
Answer. Corrected. Thank you.
b. Sometimes the information on the figures are not described (or it is not immediately visible) for instance, in Fig.1, Transmittance spectrum is given with green and red dashed lines defining something, which is not very clear what. In Fig. 2, Sample A and B are given but their difference becomes clear only after careful reading and it is not immediate. I realize that the authors put the scale, but it feels like only belongs to Sample B.
Answer. Thank you. Improved.
c. In the text, authors mentioned that they fit the Transmittance with Eq.5 (Last paragraph of page 8), while in the Figure4 caption, it is mentioned that it was Eq.6 (which is probably the one).
Answer. Thank you for showing the overlooked mistake. Corrected.
* * *
Extra Fig 1 was created to introduce into the fabrication protocol of the samples. Additional text marked in blue was inserted in Introduction section: “We use a route for obtained optical-grade 3D inorganic-structures employing ultrafast laser direct writing multi-photon lithography technique followed by high-temperature annealing. The sequence of the procedure is visually depicted inf Figure. 1”. We believe it gives more clarity regarding the motivation and preparation of the 3D crystalline-ceramics glass microstructures.

---

## Editorial Decision

published